# Extracellular Matrix Biomarkers in Colorectal Cancer

**DOI:** 10.3390/ijms22179185

**Published:** 2021-08-25

**Authors:** Min-Seob Kim, Se-Eun Ha, Moxin Wu, Hannah Zogg, Charles F. Ronkon, Moon-Young Lee, Seungil Ro

**Affiliations:** 1Department of Physiology, Digestive Disease Research Institute and Institute of Wonkwang Medical Science, School of Medicine, Wonkwang University, Iksan 54538, Korea; 1@wku.ac.kr (M.-S.K.); wumoxin@jju.edu.cn (M.W.); 2Department of Physiology and Cell Biology, Reno School of Medicine, University of Nevada, Reno, NV 89557, USA; seeunh@medicine.nevada.edu (S.-E.H.); hannahzogg@nevada.unr.edu (H.Z.); ronkon@med.unr.edu (C.F.R.); 3Department of Medical Laboratory, Affiliated Hospital of Jiujiang University, Jiujiang 332000, China

**Keywords:** colorectal cancer, extracellular matrix, collagen, metalloproteinase, thrombospondin

## Abstract

The cellular microenvironment composition and changes therein play an extremely important role in cancer development. Changes in the extracellular matrix (ECM), which constitutes a majority of the tumor stroma, significantly contribute to the development of the tumor microenvironment. These alterations within the ECM and formation of the tumor microenvironment ultimately lead to tumor development, invasion, and metastasis. The ECM is composed of various molecules such as collagen, elastin, laminin, fibronectin, and the MMPs that cleave these protein fibers and play a central role in tissue remodeling. When healthy cells undergo an insult like DNA damage and become cancerous, if the ECM does not support these neoplastic cells, further development, invasion, and metastasis fail to occur. Therefore, ECM-related cancer research is indispensable, and ECM components can be useful biomarkers as well as therapeutic targets. Colorectal cancer specifically, is also affected by the ECM and many studies have been conducted to unravel the complex association between the two. Here we summarize the importance of several ECM components in colorectal cancer as well as their potential roles as biomarkers.

## 1. Introduction

During everyday physiologic cellular division, cancer cells can be repeatedly produced [1,2]; however, homeostasis is usually preserved by a healthy microenvironment, which prevents these cells from dividing further. However, with the development of a tumor microenvironment (TME) consisting of tumor vasculature, connective tissue, and infiltrating immune cells [3], cancer cells can then infiltrate, disseminate, and ultimately become metastatic [4]. The extracellular matrix (ECM) is formed by numerous proteins, among which proteoglycan, collagen, and laminin are the main components. They provide structural support to cells, regulate cell signaling, and ultimately provide a functional platform for cell phenotypes [5,6]. Additionally, numerous types of cell surface receptors translate signals contained in the matrix and create dynamic interactions between the matrix and cells [6,7]. ECM, which constitutes a majority of the tumor stroma, is also known to play a leading role in the progression of various cancers, including colorectal cancer (CRC), and therefore promotes an environment allowing for metastasis [8]. Elevated deposition of ECM proteins in the tumor microenvironment increases the stiffness of the ECM, affecting cellular functions such as proliferation, adhesion, migration, and invasion [9,10]. Various components construct the ECM, such as collagen, elastin, laminin, fibronectin, and modulators such as metalloproteinase (MMP) that cleave the ECM components and play a central role in tissue remodeling [11]. Studies have shown that these ECM components have a significant role in both promoting the progression or metastasis of CRC and have potential clinical relevance as biomarkers [12] and therapeutic targets [3].

Collagen is a major component of the ECM and can increase tumor proliferation when normal structures are destroyed [13,14]. Tumor proliferation changes the function of collagen and is dependent on increased collagen expression [15]. After collagen remodeling, domains containing numerous hidden carcinogens within the now exposed collagen were identified, which subsequently promoted tumor metastasis [16]. Collagen remodeling is induced by collagenases, and representative collagenases are the MMP and lysine oxidase (LOX) families [17]. MMP and LOX promote collagen degradation and cross-linking, respectively [18,19]. Recently, we found increased levels of THBS4, an ECM protein, is highly associated with PDGFRβ expression in tumor tissues when compared to normal tissues of colon cancer patients [20].

Lately, there have been several reports regarding the importance of the tumor microenvironment, including that of the ECM, for therapeutic and diagnostic biomarkers [12,21]; however, the use of ECM biomarkers in CRCs has yet to be reviewed. Herein, we summarized several vital ECM proteins and representative enzymes responsible for development, remodeling, rearrangement, and manipulation of cellular functions such as MMPs, ADAMs, and LOXs, and their roles in both CRC development and progression. We also introduced the THBS family, which has potential as novel biomarkers.

## 2. Collagen

Collagen can be altered by cancer cells to regulate in situ neoplasia, invasion, and ultimately dissemination of metastatic disease [22]. Collagen works in the TME to reduce the production of chemokines, which lead to inhibition of the anti-tumor immune response. Chen et al., reported that type 1 collagen regulates IL-18, and the cancer is allowed to grow more rapidly [23]. As the tumor progresses, it is accompanied by abnormal remodeling of collagen [24,25,26], which primarily leads to excessive accumulation, altered proportions, and arrangement of collagen [27,28,29]. Liang et al. analyzed the association between the collagen related genes and the incidence and prognosis of CRC using biometric databases and demonstrated that the overall survival (OS) time was worse in patients with upregulated expression of the gene combination for collagen and collagenase. Based on these results, they reported that the coding genes COL1A1-2, COL3A1, COL4A3, COL4A6, and MMP2 could be used as biomarkers to predict the prognosis of patients with CRC [17]. Additionally, collagen deposition is characteristic of CRC, and collagen types I, VI, VII, VIII, X, XI, and XVIII were accumulated in CRC samples [30,31,32,33,34,35,36].

Zhang et al. demonstrated that the COL1A1 expression was significantly upregulated in CRC tissues and cell lines with both wild type and mutant KRAS [37]. Further, increased COL1A1 expression in CRC was significantly associated with serosal invasion, lymphatic involvement, and hematogenous metastasis [37]. Moreover, it was found that COL1A1 regulates the WNT/PCP pathway to promote metastasis in CRC and that inhibition of COL1A1 can suppress CRC cell migration [38].

Type 1 collagen was found to be increased in tumor tissues compared to normal tissues, and its mRNA was increased in the blood of CRC patients compared to healthy individuals [39,40]. Additionally, expression of collagen type I, III, and IV turnover products were significantly increased in the serum of patients with CRC, especially in the later stages [41,42]. Moreover, type I and type IV collagen were upregulated in the urine [43,44] and plasma [45] of patients with colorectal liver metastasis (CLM), respectively. These results suggest that not only type I, but also type III and IV collagen may have clinical relevance in CRC. Further, the circulating type IV collagen levels were strongly increased in the presence of CLM [46]. In addition, it has been indicated that the α1 and α2 chains of type IV collagen are major components seen in the desmoplastic reaction of CLM [46]. Patients with CLM were found to have an 81% and 56% increase in the levels of circulating COLIV and CEA, respectively, at the time of diagnosis [45]. It can therefore be concluded that COLIV is a promising tumor marker for CLM and may possibly be used to detect postoperative CLM recurrence. Additionally, the combination of COLIV and CEA is superior to either of the markers alone in detecting CLM [45].

Moreover, among the formation of collagen type III (PRO-C3), VI (PRO-C6), and degradation of collagen type VI (C6M and C6Mα3), PRO-C3 was the most effective in predicting the survival outcome of metastatic colorectal cancer (mCRC) patients regardless of other risk factors [47]. In another study, specific fragments of degraded type I, III, and IV collagen and type III collagen formation in serum showed that stage IV metastatic patients could be distinguished from all other stages [42]. Type 1 collagen in tumor tissues promotes EMT (epithelial mesenchymal transition) in CRC cells, and this process was induced by activation of PI3K/AKT/snail signaling pathway conducted by integrin α2β1 [48]. These results demonstrated that collagen can promote tumor growth and distant metastasis in CRC patients via the pathway. Type 1 collagen also downregulates E-cadherin and β-catenin and promotes the expression of stem cell markers CD133 and BMI1 [49]. This study suggests that the interaction between CRC cells and type 1 collagen results in EMT-like changes, loss of differentiation, and increased expression of stem cell markers.

COLXIA1 was found in sporadic colorectal cancer that expresses high levels of COLXIA1 mRNA, whereas there was no expression in normal tissue [50]. The authors suggest that COLXIA1 overexpression may be related to the APC/β-catenin pathway, which is dysregulated in most sporadic colorectal cancers as well as familial adenomatous polyposis [51].

COL12A1 is involved in the development of CRC in transcriptomic analysis [52]. Also, COL12A1 expression was negatively correlated with the methylation levels of the COL12A1 promoter, and hypermethylation of the gene was found in low-stage tumors and negative node metastasis [53]. Patients with higher COL12A1 expression were negatively associated with disease-free survival (DFS), indicating that COL12A1 is a poor prognostic indicator for CRC [53].

There are increased levels of the C-terminal end of type XVI collagen in the serum of CRC patients [54]. In addition, type XVI collagen is upregulated by transforming growth factor-β (TGF-β) [55]; high levels of TGF-β are associated with metastasis and a poor outcome in CRC patients [55,56]. An increase in collagen XVII expression was significantly associated with a higher TNM stage in colorectal carcinoma [57]. It is also associated with infiltrative growth patterns and tumor budding, lymph node involvement, and distant metastasis [57]. Additionally, collagen XVII gene expression is upregulated in a STAT3 dependent manner and stabilizes laminin-5 (laminin-322) [58,59]. It has been demonstrated that upregulated collagen XVII-laminin-5 mediates anoikis resistance, which plays an essential role in determining tumor initiation and metastasis in cancer stem cells [58].

Taken together, collagens promote the development and progression of CRC in various ways and provide both novel clinical treatment directions and have potential as biomarkers to aid in staging, monitoring the progression of the disease, analyzing the response to treatment, and monitoring for the recurrence of CRC.

## 3. Laminin

Laminins are large multimeric basement membrane proteins with functional units constructed through the assembly of one α, one β, and one γ isoform [60]. Laminin is involved in tumor angiogenesis, cell infiltration, metastasis, cytokines and proteases leading to tumor progression and drug resistance [61,62]. Several studies have reported the association between CRC and laminin chains and its potential as a biomarker of CRC [61,63,64].

Laminin-5 (Laminin-332) is a heterotrimer composed of α3, β3, and γ2 chains that are essential for epithelial cell migration and basement membrane attachment [59]. In the univariate and multivariate analyses of the laminin-332 γ2 chain for CRC patient survival, the prognosis was markedly reduced in patients with an increased number of immune positive tumor cells [63]. Additionally, the γ2 chain synergistically contributes to the formation of budding tumor cells with MMP-2, and the level of γ2 chain expression in the submucosal and subserosal invasive fronts was an independent prognostic factor [65]. Laminin γ2 was significantly correlated with poorer clinical outcomes such as disease-specific, recurrence-free, disease-free, and OS in CRC [66]. Further, stable overexpression of laminin γ2 promoted proliferation, migration, and invasion of CRC cells [66]. In addition, β3 chain expression was a poor prognostic factor for CRC, and laminin-332 was associated with chemical resistance to 5-FU-based chemotherapy [67].

Laminin β-1 is a glycoprotein that is not known to be secreted by colon cancer cells [68]. However, the level of laminin β-1 was significantly higher in the serum samples of CRC patients compared to the healthy controls [68,69,70,71]. Also, according to a ROC analysis, laminin β-1 performed better than CEA in distinguishing control subjects versus CRC patients [68]. Moreover, the use of laminin β-1 in combination with CEA further improved the diagnostic efficacy [68].

Abnormal methylation at the target CpG sites of Laminin α2 (LAMA2) was observed at a high frequency in CRC tumor tissues and less frequently in normal tissues adjacent to the tumor, revealing that it is also a potential biomarker [72]. Qin et al. showed that high Laminin 521 expression is a frequent feature of metastatic dissemination in CRC and promotes cell invasion and self-renewal. Furthermore, they demonstrated the ability to promote self-renewal of Biolaminin 521 LN (LN521) through the engagement of integrins α3β1 and α6β1 and activation of STAT3 signaling [64]. In addition, changes in laminin expression have been reported in various inflammatory diseases. Several studies have reported Laminin α4, α5, α6β1, α3β1, α7, and β2 are upregulated TNF-α, MIP-1β, IL-1β, IL-6, and IL-8 [73,74,75,76].

Some laminins suggest a possible prognostic approach of CRC progression and metastasis, although some studies have only been performed using chains of laminin [61,69,77]. In addition, the expression and role of laminin in CRC may be useful for the development of novel biomarkers, as well as a potential target of novel treatment modalities.

## 4. Matrix Metalloproteinases

Matrix metalloproteinases (MMPs) and tissue inhibitors of matrix metalloproteinase (TIMPs) have been known to play a role in the process of tumor invasion, progression, and metastasis of CRC [78]. Accordingly, many researchers have been unraveling the potential of a variety of MMPs and TIMPs as potential biomarkers of CRC [79,80,81,82,83]. Here we describe the value of MMP as a biomarker in CRC based on these studies.

MMP-1 is expressed in various cells, including fibroblasts, hepatocytes, and tumor cells, and the level of expression is consistently high even without evident stimulation [84]. MMP-1 is described in a variety of advanced cancers [85,86,87], and colon cancer is no exception [88]. Upon analysis of the serum MMP-1 activity before surgical resection in 75 patients with CRC [78,89], it was clear that MMP-1 activity may predict the future progression of malignant cells. These results indicate that MMP-1, activated in primary lesions and found in serum, may be helpful for the clinical diagnosis of CRC [90]. Expression of MMP-1 and MMP-7 was elevated in patients with primary colorectal cancer with or without liver metastasis [91,92]. These results indicated that the activity of these proteases is pivotal for metastasis, and positive MMP-1 expression in primary colorectal tumor tissue was a significant predictor of liver metastasis [79,84,88,93]. They speculated that the prognostic impact of protein marker expression in terms of intrahepatic recurrence would be insignificant [79]. Additionally, MMP-1 is independently associated with cancer-specific survival (CSS) in patients with CRC, and MMP-1 expression in tumor-free mucosa can be used to help guide providers in the identification of patients with colorectal cancer who may require additional systemic chemotherapy or more intensive adjuvant treatments in addition to curative resection [93].

MMP-2 and MMP-9 are known as type IV collagenases and are associated with CRC progression, angiogenesis, and metastasis [94]. These proteins can be found in serum and feces and may have potential as biomarkers as they can distinguish between CRC patients and healthy controls [95]. MMP-2 and MMP-9 were increased in the serum of colorectal cancer patients, and their diagnostic sensitivity was higher than that of other biomarkers currently used in clinical practice, such as CEA and CA19-9 [95]. However, serum MMP-9 showed results that could not be considered beneficial for the diagnosis of advanced neoplasia in the CRC family-risk population screening [96]. Also, levels of MMP-9 were measured in stool samples from 125 colon cancer patients and showed that MMP-9 may be a promising noninvasive marker of CRC [97]. Increased expression of MMP-2 and MMP-9 in normal tumor-free mucosa adjacent to colorectal tumors was independently associated with poor prognosis in colorectal cancer patients [98]. Additionally, it has been found that TIMP-2 can regulate MMP-9 to predict the prognosis of patients with CRC [82]. Wang et al. demonstrated that TIMP-2 inhibits cell invasion and migration by regulating MMP-9 at the mRNA or protein level and may be more effective than TIMP-2 or MMP-9 alone to predict the prognosis of patients with CRC [82].

MMP-7 also exhibits proteolytic activity on components of the extracellular matrix, is often overexpressed in human cancer tissues and is associated with cancer progression [91]. Overexpression of MMP-7 is observed in ~80% of CRC [99], and serum levels of MMP-7 are associated with cancer progression and decreased survival in advanced CRC [100]. Also, MMP-7 promotes cancer invasion through proteolytic cleavage of ECM proteins and by activating other MMPs, including proMMP-2 and proMMP-9 [91]. Despite these characteristics of MMP-7 in cancer, clinical use of MMP-7 inhibitors has been disappointing due to its poor therapeutic effect and side effects [101]. However, the association between MMP-7 tissue expression and prognosis in colorectal cancer was evaluated, and it was shown that MMP-7 expression has the potential to be a prognostic marker for a poor 5-year outcome in colorectal cancer [92]. Additionally, serum levels of TIMP-1 and MMP-7 were significantly higher in CRC patients than in healthy controls [80]. These two factors were more sensitive than CA19-9 [80], suggesting that they could be effective biomarkers in patients with metastatic CRC with good sensitivity. Their data show that TIMP-1 and MMP-7 correlate with liver involvement in CRC patients. Additionally, these entry levels are prognostic factors that correlate with OS [80]. Similarly, as a result of performing a meta-analysis on MMP-7, it was demonstrated that overexpression of MMP-7 is associated with poor OS, poor DFS, and reduced 5-year survival rate in CRC patients [102].

MMP-13 is also associated with the development and progression of colon cancer [83,103]. As a result of evaluating MMP-13 mRNA expression in cancer tissues of CRC patients by the membrane array method, the stage of cancer progression is significantly associated with MMP-13 overexpression, and it was confirmed that patients with MMP-13 overexpression had a 7.989-times higher risk of recurrence after surgical resection compared with patients without MMP-13 overexpression [104]. Therefore, these MMP-13 studies show that MMP-13 has significant potential as a CRC prognostic marker with several reports that high levels of MMP-13 are associated with liver metastasis, poor prognosis, and early recurrence [83,103]. Additionally, the immunoreactive score (IRS) of MMP-13 was suitable for evaluating the pathologic grade of precancerous and cancerous colorectal lesions, showing that MMP-13 can be applied to stratification of prognosis as an excellent marker of CRC [105].

MMP-14 has been reported to promote invasion and metastasis in several cancers, and several studies have been conducted as a biomarker candidate in colorectal cancer as well. The mRNA and protein levels of MMP-14 were increased in CRC tissues compared to normal tissues, and patients with higher expression had lower 5-year DFS and OS than those with lower expression [106]. Another study reported increased expression of α5β1-integrin and MMP-14 in patients with poor histological differentiation, lymph node metastasis, and high clinical stage of CRC [107]. Interestingly, MMP-14 and α5β1-integrin have the common functions of activating MMP-2 [108,109,110,111] and stimulating tumor angiogenesis [112,113]. However, there is also a study in which no correlation was observed between MMP-14 expression and the prognosis of CRC [114]. Collectively, these studies suggest that MMP-14 may be a biomarker in CRC, but further studies are needed.

Numerous studies have proven the outstanding value and utility of MMP in CRC. These studies suggest that MMPs contribute to the progression and metastasis of CRC and lead to a poor prognosis. Naturally, it is still insufficient to determine that this is an ideal biomarker. Further research is warranted to test the efficacy of MMP as a biomarker for CRC.

## 5. LOX

The lysyl oxidase (LOX) family is a member of secreted copper-dependent amine oxidases that play an essential role in mediating ECM remodeling and cross-linking of collagen and elastin [115]. The LOX family is composed of LOX and four LOX-like proteins (LOXL1, LOXL2, LOXL3, and LOXL4) [116], which catalyze the conversion of lysine residues into highly reactive aldehydes and mediate cell adhesion, invasion, and motility [117].

Baker et al. [118] revealed that LOX was significantly elevated in tumor tissues, and the increase was more significant in metastatic tumors, suggesting its role in tumor progression. In vitro and in vivo models showed LOX stimulates endothelial cell division and drives tumor angiogenesis through the activation of the Akt-VEGF pathway in CRC [119]. P-selectin-mediated platelet aggregation increases LOX expression, causing tumor ECM remodeling and stiffening, thereby promoting the progression of CRC [115]. Intracellular LOX is significantly associated with poor survival and a prognostic biomarker for lung/hepatic metastasis of colon cancer [120]. However, Csiszar et al. demonstrated that LOX mRNA expression was reduced in CRC patients, indicating that LOX had the tumor-inhibiting effect [121]. LOXL1 acts as a critical tumor suppressor in regulating tumor growth, invasion, and metastasis by inhibiting the activity of yes-associated protein (YAP) [122]. LOXL2 is involved in the occurrence of CRC invasion and metastasis and can be an independent prognostic biomarker [123,124,125]. Moreover, LOXL2 knockdown attenuates the proliferation and migration of CRC cells and induces cell cycle arrest and apoptosis in vivo and in vitro [126].

Additionally, in a meta-analysis study, Zhang et al. [127] revealed that increased LOXL2 expression was obviously associated with worse clinical outcomes and might serve as a prognostic biomarker in digestive system cancer. Palmieri et al. [128] found that the expression of LOXL4 in neutrophils was a potential surrogate biomarker for the subtype of colorectal cancer with liver metastasis.

In summary, the LOX family is associated with the regulation of colon cancer cell proliferation, migration, invasion, and metastasis and might be a potentially effective molecular marker for diagnosis and treatment of CRC.

## 6. ADAM

A disintegrin and metalloproteinase (ADAM) is the metzincin proteolytic enzyme superfamily of zinc-based metalloproteinase, which regulates the shedding of membrane-bound proteins, cytokines, growth factors, ligands, and receptors [129].

ADAM8 is overexpressed in CRC tissues and is associated with worse OS and DFS, which can serve as a prognostic biomarker for CRC patients [130]. miR-20b reduces 5-FU resistance and induces apoptosis of colon cancer cells by suppressing ADAM9/EGFR [131]. ADAM12 overexpression enhances CRC proliferation and inhibits cell apoptosis, which is related to cancer staging, distant metastasis, and poor prognosis [132]. In addition, Wang et al. [133] found that ADAM12 showed promoter hyper-methylation and lower expression in CRC tissues compared to adjacent normal tissues. Recent studies have also suggested that ADAM15 is overexpressed in several types of cancer and is involved in metastatic tumor progression [134,135,136,137,138,139,140]. In CRC, Claudin-1 expression was increased through the ADAM15 pathway and was associated with the progression of CRC and metastasis [134,141]. However, Toquet et al., reported that ADAM15 expression is decreased in advanced colon cancer and this reduction is associated with metastasis and shorter overall survival [142]. Compared with normal colonic mucosa, ADAM17 is overexpressed in primary and metastatic CRC tissues, and a selective ADAM17 inhibitor leads to concentration-dependent decreases in cell proliferation and activity [143]. Chemotherapy activates ADAM17 expression, leading to growth factor shedding, growth factor receptor activation, and drug resistance in CRC [144]. Pharmacological inhibition or siRNA silencing of ADAM17 activity, combined with chemotherapy, has potential therapeutic benefits for CRC patients [144,145,146,147,148,149]. Wang et al. [150] suggested that Nox1 promoted CRC metastasis through the regulation of ADAM17 stability. ADAM17 can serve as a prognostic biomarker for advanced CRC, contributing to the development of new therapies focused on reducing tumor metastasis [81]. ADAM transcription is activated by the Wnt signaling pathway and can lead to the recruitment of epigenetic complexes that promote CRC metastasis [151]. 2-Deoxy-D-glucose and siRNA suppresses the activation of ADAM10 and ADAM17, down-regulates mesenchymal properties, reduces the secretion of EMT-associated cytokine, and renders the tumor susceptible to anti-cancer drug treatment [152,153,154].

Collectively, ADAM may serve as a promising biomarker and therapeutic target for CRC patients, and further investigation is needed.

## 7. Proteoglycan

Proteoglycans (PGs) are composite molecules in which the protein core is covalently linked to glycosaminoglycans chains (GAGs). Major GAGs include heparan sulfate, chondroitin sulfate, dermatan sulfate, hyaluronan, and keratan sulfate. According to cell location, all PGs can be divided into intracellular, pericellular, ECM proper, and extracellular PGs. PGs have an essential role in maintaining ECM structure and are related to cancer pathogenesis [155].

Each malignant tumor has a unique PG profile, which is closely related to its biological behavior and differentiation. Chondroitin sulfate proteoglycan serglycin is a poor prognostic biomarker, which acts as an essential downstream target of HIF-1α to promote CRC metastasis [156]. Serglycin is a unique intracellular PG that is constitutively expressed and secreted at high levels in the culture medium of aggressive colon cancer cells [157].

Loss of heparin sulfate proteoglycan syndecan-1 (Sdc-1) expression is related to histological differentiation and clinical staging in CRC patients [158]. Sdc-1 is involved in chemotherapy resistance through the EGFR pathway and may be a new prognostic biomarker for CRC [159]. Depletion of Sdc-1 is associated with activation of integrins and focal adhesion kinases, which then generate signals for enhanced aggressiveness and cancer stem cell properties [160]. In addition, sdc-1 suppresses cell growth and migration via blocking Ras/Raf/MEK/ERK and JAK1/STAT3 pathways in human CRC cells [161]. Syndecan-2 (Sdc-2) exerts carcinogenic effects through the activation of the EMT and MAPK pathways, as well as the interaction with the ECM that is produced by stromal fibroblasts in CRC [162,163]. Moreover, fecal Sdc-2 methylation levels measured by linear target enrichment (LTE)-quantitative methylation-specific real-time PCR can be used as a biomarker for noninvasive detection of CRC patients [164,165,166].

SLRPs, a family of proteoglycans that share a common leucine-rich-repeat proteoglycans with strongly involved in modulating the tissue hydration and assembly of fibrillary collagen, including biglycan, decorin, fibromodulin, and lumican, can affect fibril growth rate, size, morphology, and content. As an SLRPs family, biglycan is significantly higher in CRC tissues than that of corresponding normal tissues, which is related to poor tumor differentiation, lymph node metastasis, and distant metastasis [167]. In addition, biglycan can directly increase VEGFA expression in colon cancer cells, thereby promoting tumor angiogenesis and cancer growth [168]. Another member of the SLRPs family, decorin, can inhibit EMT and CRC metastasis through interaction with E-cadherin [169,170]. Finally, Radwanska et al. found that enhanced lumican expression and its presence in ECM have an impact on colon cancer cell migration through up-regulation of gelsolin and filamentous actin reorganization [171]. Lumican inhibits SNAIL-induced melanoma cell migration in vitro and vivo specifically by blocking MMP-14 activity [172]. Furthermore, the expression of lumican is increased during colorectal adenoma-to-carcinoma progression and able to predict good clinical outcomes for stage II and III colon cancer, [173,174]. There is a key role of PG in the pathogenesis of CRC, and elucidating the changes in PG expression, structure, and location may provide insights for the development of innovative biomarkers as well as selective and more effective therapies.

## 8. Thrombospondin

Thrombospondin (THBS) family, a member of secreted Ca^2+^-binding ECM glycoproteins that share a highly conserved C-terminal region, are composed of five homologous members and can be divided into two groups [175]. Group B (THBS3, THBS4, and THBS5) has fewer domains than group A (THBS1 and THBS2) and is involved in embryonic development [176,177] and in skeletal growth [178]. However, group A has unique N-terminal domains and is more related to cardiovascular development [177,179]. THBS serve important roles in numerous physical and pathological processes, such as cell differentiation, proliferation, migration, fibroblast apoptosis, vascular homeostasis, immunity, and wound healing [180] as well as glucose and insulin metabolism [181].

Additionally, THBS have long been associated with the regulation of angiogenesis and cancer by regulating multiple physiological processes that determine cancer growth and spreading (angiogenesis, inflammation, metabolic changes, and other properties of the ECM) [181]. Despite several studies that have reported the relationship between THBS expression and cancer development in different types of cancers [182,183,184], only a few studies have researched thrombospondins in colorectal cancers. THBS1 was the first member to be identified among the thrombospondin family and has been mostly studied focusing on the development and metastasis in several cancers including colorectal cancer as an anti-angiogenic factor in the tumor microenvironment [180,183,184,185,186,187,188].

There have been several reports that there is a protective role of THBS1 in the anti-angiogenic and anti-tumorigenesis in colorectal cancer [182]. For example, THBS1 is mainly expressed in fibroblasts in the tumor stroma and there is better prognosis with activation of TGFβ-1 in CRC [186]. In another study, THBS1 was found to be highly expressed in the normal colonic mucosa, but was progressively lost depending on the adenoma size and even completely absent in the epithelial component of carcinomas [183]. This suggests that Wnt signaling, rather than K-ras, has a role in repression of THBS1 gene expression [185]. Under a low (5%) fat diet, Apc^Min/+^:THBS1^−/−^ mice showed lower survival and higher tumor multiplicities in the small and large intestine relative to Apc^Min/+^ mice [187]. Additionally, inhibition of THBS1 by miR-194 promoted angiogenesis and tumor growth of colonic carcinoma xenografts [189]. Conversely, THBS1 showed a significant correlation with poor survival in resected colorectal liver metastases [188].

However, the expression pattern of THBS1 and THBS2 and their roles in CRC have provided several controversial results. Tokunaga et al. showed that THBS2 in CRC expressed a significantly lower incidence of hepatic metastasis, whereas THBS1 expression had no apparent correlation [190]. Differing from these results, Yosida et al. reported that THBS2, regarded as an angiostatic factor, was significantly increased in CRC but not THBS1 [191]. In the opposite way to THBS1 regulation by miR-194, THBS2 has a binding domain in its 3’-untranslated region for miR-203a-3p, and downregulation of THBS2 by miR203a-3p inhibits CRC progression and metastasis [192]. There were also ambiguous results in terms of THBS2 expression such that low THBS2 expression reflected poor prognostic factors [193], while there was no correlation with improvement in metastatic colorectal cancer [194].

The role of THBS3 and THBS5 in the regulation of cancer growth has only been studied once with respect to THBS3 in osteosarcoma [195]. However, THBS4 has been suggested as a valuable gene and regulator and has been found to target several cancers including, breast [196,197], gastric [198,199], and hepatocellular carcinoma [200].

Most studies have shown higher THBS4 gene expression in gastric cancer [201], breast lobular carcinomas [197], diffuse type of gastric adenocarcinoma [198], HCC [202], and cancerous ovarian and renal cultures [203]. The remarkable activation of THBS4 expression in tumors is most likely regulated through the interactions of invading tumor cells with stromal fibroblasts in the local microenvironment [196].

Recently, it was found that there are increased levels of THBS4 in colorectal cancer patients and it is highly associated with PDGFRβ expression in tumor tissues compared to normal tissues [20]. Further, the expression level of THBS4 was also significantly increased with increased expression of PDGFRβ, despite no significant changes in mRNA [20]. This may implicate post-translational modification of THBS4. Additionally, TGF-β and PDGF-D were involved in this pathway and promoted the proliferation, migration, and adhesion of a colonic fibroblast, and fibroblasts can actively diffuse around and inside a neoplastic mass favored by the action of these factors [204,205]. Although one controversial study suggested that THBS4 may act as a tumor suppressor gene in colorectal cancer [206], considering all of the results in the other types of cancer [185,196,199,201], higher expression of THBS4 is more likely associated with tumor development. However, how transcriptional and post-transcriptional levels of THBS4 are regulated in by interactions of tumor cells and stromal fibroblasts involved remain elusive.

According to the THBS4 expression data in the colon adenocarcinoma (COAD) in the Cancer Genome Atlas (TCGA), although it showed THBS4 is not a prognostic marker in COAD, there seems to be a strong relationship between stage of disease, expression levels and/or survival rate and survival rate is significantly lower in the higher expression individuals [207]. Interestingly, there also seems to be a gender difference in the relationship between THBS4 expression and survival rate because females showed a significantly lower survival rate than males [207].

Collectively, THBS family is associated with the development and progression of colorectal cancers. Further studies on THBS are required to explore possible diagnostic and therapeutic approaches in CRC.

## 9. Conclusions

Many molecular markers are associated with the occurrence, progression, and prognosis of carcinoma. There are seven ECM biomarkers for CRC as shown in Table 1. Additionally, Figure 1 summarizes the differential expression of ECM biomarkers in CRC. Elevated ECM proteins in the tumor microenvironment increase the stiffness of the ECM, affecting cellular functions such as proliferation, adhesion, migration, tissue remodeling, and regulation of immune system [9,10,11,208].

Collagen has important implications in the structural growth of tissue and regulates molecular signaling processes. A variety of tumors have been shown to overexpress collagen and its proteolytic components have been linked to increased invasiveness of tumors [41,84]. For example, collagen type I, III, and IV are upregulated by increased expression of TGF-β and this increase is associated with metastasis in CRC [27,55]. MMP-2 and MMP-9 are known as type IV collagenase and are increased in CRC patients and are associated with colorectal cancer progression [95]. MMPs are key promoters of cancer progression, CRC migration, and metastasis in CRC. MMP activity is regulated by TIMPs; hence dysregulated MMP–TIMP expression may favor proteolysis, thereby contributing to an environment catalyzing metastasis [175,219]. Many ECM remodeling enzymes including MMPs and *LOX family* oxidases are expressed during malignant transformation, progression, and metastasis of CRC [116]; increased MMPs and LOX expression in CRC correlate with advanced disease progression and a poor prognosis [40,93,94,97,98,99,104].

ADAM, PGs, and THBS are responsible for the regulation of cell differentiation, proliferation, migration, and fibroblast apoptosis [129,155,180]. Overexpression of ADAM in CRC enhances cell proliferation and inhibits apoptosis, which is related to cancer staging, distant metastasis, and poor prognosis [132]. However, in some reports, decreased expression of ADAM was observed in different types of carcinomas. For example, even though one study suggested that ADAM15 is decreased in CRC [142], there are several other studies that found there is an increase in expression of ADAM15 in several types of cancer [134,135,136,137,138,139]. However, one study has identified an increase in the downstream gene of ADAM15 in CRC [134]. It is suggested that the role of ADAM15 in cancer progression is tissue-specific. Although most studies have shown higher THBS gene expression in different types of cancer, the exact mechanism linking the THBS family to these cancers is not known.

Given the various roles of ECM biomarkers in the tumorigenesis of CRC and other cancers, it is a promising target for pharmaceutical intervention. Increased expression of ECM biomarkers is associated with tumor invasion, metastasis, and a poor clinical outcome for cancer patients [8]. Furthermore, while many studies have demonstrated overexpression of ECM proteins, it is not known how they may regulate tumor invasion and metastasis. Further, cells expressing high levels of ECM proteins have an increased capacity to proliferate, invade, and metastasize as a consequence of this increased expression. This suggests that inhibition of ECM proteins may provide a novel and effective treatment option for patients with CRC to prevent metastatic progression. Moreover, future therapies targeting the ECM also have potential use in prevention of precancerous lesion progression based on the profound role of the ECM in tumor development, progression, and ultimately metastasis. In turn, further elucidation of these intricate mechanisms within various cancers likely has profound implications; especially in the personalized medicine realm with tumor specific targeted therapies leading to vastly more efficacious treatments.

## Figures and Tables

**Figure 1 ijms-22-09185-f001:**
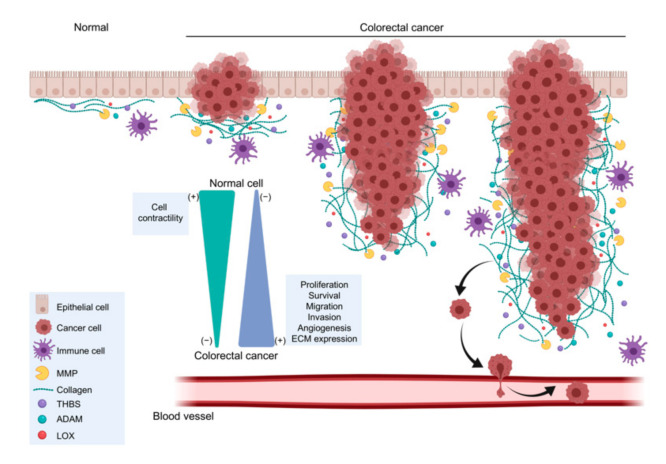
A simplified diagram of extracellular matrix biomarkers differentially expressed in colorectal cancer and normal tissue. MMP, Matrix Metalloproteinase; THBS, Thrombospondin; ADAM, A disintegrin and metalloproteinase; LOX, Lysyl oxidase. Adapted from “Melanoma Staging” and “Tumor Cell Metastasis”, by BioRender.com (2020). Retrieved from https://app.biorender.com/biorender-templates (accessed on 22 July 2021).

**Table 1 ijms-22-09185-t001:** Extracellular matrix biomarkers in cancer.

Name	Function	Gene Name	Expression (Verse Normal Tissue/Cells)	Tumor Type	Reference
Collagen	Cell adhesion	COL1A1, COL3A2, COL4A3, COL4A6, PRO-C3, PRO-C6, C6M, C6Mα3, ColXIA1, COL12A1	Upregulated	Tumor stroma, lung tumor, thyroid cancer, colorectal cancer, osteosarcoma, breast cancer, hepatocellular carcinoma, ovarian cancer	[17,37,38,39,40,41,42,47,52,53,209]
Laminin	Tumor angiogenesis, cell infiltration, metastasis, drug resistance	LAMA2, Laminin β-1, Laminin γ2	Upregulated	Colorectal cancer, pancreatic adenocarcinoma, lung adenocarcinoma	[60,63,65,210]
Matrix Metalloproteinase (MMP)	tumor invasion, progression, metastasis	MMP-1, MMP-2, MMP-7, MMP-9, MMP-13, TIMP-1, TIMP-2	Upregulated	Colorectal cancer, gastric cancer, breast cancer, lung tumor	[83,84,85,91,95,97]
Lysyl oxidase (LOX)	ECM remodeling, tumor growth	LOXL1, LOXL2, LOXL3 LOXL4	Upregulated	Hepatocellular cancer, colorectal cancer, lung tumor	[211,212]
A disintegrin and metalloproteinase (ADAM)	Regulation of cytokines and growth factors, cell proliferation	ADAM-8, ADAM-10, ADAM-12, ADAM-15	Upregulated: ADAM-8, ADAM-12, ADAM-15 (lung tumor, pancreatic cancer) Downregulated: ADAM-10, ADAM-15 (Colorectal cancer)	Colorectal cancer, lung tumor, pancreatic cancer	[129,133,142,144,145,148,151,152,213]
Proteoglycan (PG)	Enhancement of cell viability, cell proliferation, invasion, metastasis, regulation of cytokine, cell adhesion and migration, angiogenesis	Serglycin, Glypican (GPC)-1, GPC-4, GPC-5, Syndecan (SDC)-1, SDC-2, SDC-3, SDC-4, HSPG2	Upregulated: Serglycin, GPC-1, GPC-4, SDC-1 (Pancreatic cancer, breast cancer), SDC-2, SDC-3, HSPG2 Downregulated: SDC-1 (Colorectal cancer), SDC-4, GPC-5,	Nasopharyngeal cancer, glioblastoma, hepatocellular cancer, colorectal cancer, pancreatic cancer	[155,160,161,214,215,216,217,218]
Thrombospondin (THBS)	Cell proliferation, carcinogenesis, fibroblast apoptosis, vascular homeostasis	THBS1, THBS2, THBS3, THBS4, THBS5	Upregulated	Lung tumor, thyroid cancer, colorectal cancer, osteosarcoma, breast cancer, hepatocellular carcinoma	[176,179,180,181,182,183,184,185,186,187,188,189,190,191,192,193,195,196,198,199,200,201,202,206]

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
