# Peer review of "Extracellular Matrix Biomarkers in Colorectal Cancer"

_ijms, 2021, doi:10.3390/ijms22179185_

Round 1

Reviewer 1 Report

My major concern about this review article is the reason why the authors chose these particular biomarkers.

The biomarkers presented are well described but why these biomarkers? Why not other proteoglycans, for example SLRPs? It has to be really explained why authors chose to devote the article in fibronectin, collagen, etc.

The introduction is quite fair. 

ECM has to be well described (the structure, the molecules comprising it, the interactions between cells- matrix, the reason why in cancer research we study the signaling cascade of ECM). You can cite the bible of ECM, the book  ''the biology of ECM'' by Nikos Karamanos.

Moreover, the stages of cancer development is fairly described (intravasation, extravasation, e.t.c.). I would like to see more detail in it.

The paragraph about collagen is quite well organized, although it is more clinically oriented, and I would like to see more literature about signaling. For example, I would to see how collagen is related with other ECM molecules, such as lumican. You can see the bibliography of Dr. Brezillon. 

Regarding MMPs, MMP-14 is not cited and it the most studied in cancer research. In melanoma for example, MMP-14 is well studied and melanoma with colon cancer are quite close ''Lumican inhibits in vivo melanoma metastasis by altering matrix-effectors and invadopodia markers'', ''Small leucine-rich proteoglycans and matrix metalloproteinase-14: Key partners?''.

minor points: in vitro, in vivo should be in italics 

COLIV, CEA should be explained 

row 303: serglycin is  a unique (check grammar errors very carefully)

In the paragraph of SLRPS, lumina has to be cited. I give you an example of literature ''Lumican inhibits SNAIL-induced melanoma cell migration''. specifically by blocking MMP-14 activity

Serglycin is also fairly described. 

Moreover, in the table, PGs are presented with just few functions. There are so many more. Please flourish it with adequate publications.

The analysis of the figure is fair, please improve it.

The letter font is not the same everywhere. How can an article be submitted without check of the letter font. Also, some are in italics, some are centralized.        

Author Response

Reviewer 1

My major concern about this review article is the reason why the authors chose these particular biomarkers. The biomarkers presented are well described but why these biomarkers? Why not other proteoglycans, for example SLRPs? It has to be really explained why authors chose to devote the article in fibronectin, collagen, etc.

Response: We originally focused on collagen, proteoglycan, and laminin as biomarkers in CRC in this review. We added the SLRPs and other proteoglycans as suggested.

See rows 354-371 in pages 7-8.

The introduction is quite fair. ECM has to be well described (the structure, the molecules comprising it, the interactions between cells- matrix, the reason why in cancer research we study the signaling cascade of ECM). You can cite the bible of ECM, the book ''the biology of ECM'' by Nikos Karamanos. Moreover, the stages of cancer development is fairly described (intravasation, extravasation, e.t.c.). I would like to see more detail in it.

Response: We cited the represented reference and updated the manuscript with more detailed molecular mechanisms.

The paragraph about collagen is quite well organized, although it is more clinically oriented, and I would like to see more literature about signaling. For example, I would to see how collagen is related with other ECM molecules, such as lumican. You can see the bibliography of Dr. Brezillon. 

Response: We added collagen and proteoglycan pathways.

See rows 106-113, 342-353 in pages 3 and 7.

Regarding MMPs, MMP-14 is not cited and it the most studied in cancer research. In melanoma for example, MMP-14 is well studied and melanoma with colon cancer are quite close ''Lumican inhibits in vivo melanoma metastasis by altering matrix-effectors and invadopodia markers'', ''Small leucine-rich proteoglycans and matrix metalloproteinase-14: Key partners?''.

Response: We fixed it.

Minor points: in vitro, in vivo should be in italics 

Response: We fixed it.

COLIV, CEA should be explained 

row 303: serglycin is a unique (check grammar errors very carefully)

Response: We fixed it.

In the paragraph of SLRPS, lumina has to be cited. I give you an example of literature ''Lumican inhibits SNAIL-induced melanoma cell migration''. specifically by blocking MMP-14 activity

Serglycin is also fairly described. 

Response: We added 8 more references.

Moreover, in the table, PGs are presented with just few functions. There are so many more. Please flourish it with adequate publications.

Response: We fixed it.

The analysis of the figure is fair, please improve it.

The letter font is not the same everywhere. How can an article be submitted without check of the letter font. Also, some are in italics, some are centralized.

Response: We redrew and replaced the figure.

Reviewer 2 Report

In perspective, there are possible applications in the general oncological practice? Besides, there is any impact on the immunity system?  About this you can read the articles listed below.  The text is quite long and, if possible, some other explanatory could be useful. 

  • Roncati L, Manenti A, Barbolini G, Maiorana A. Deep inside of gastric signet-ring cell carcinoma. Neoplasma. 2018;65(4):579-584. doi: 10.4149/neo_2018_170404N246. PMID: 30064231.
  • Roncati L, Manenti A, Piscioli F, Pusiol T, Barbolini G. Immunoscoring the lymphocytic infiltration in carcinoid tumours. Histopathology. 2017 Jun;70(7):1175-1177. doi: 10.1111/his.13168. Epub 2017 Mar 10. PMID: 28116775.
  • Roncati L, Gasparri P, Gallo G, Bernardelli G, Zanelli G, Manenti A. Appendix tumor  In  A. Birbrair  (ed.) Tumor microenviroments in organs,  Advances in Experimental  Medicine and Biology 1226; Springer Nature Switzerland  AG 2020. https://.org/10.1007/978-3-030-36214-0_7

Author Response

Reviewer 2

In perspective, there are possible applications in the general oncological practice?

Response: ECM is a potentially useful biomarker and target for several clinical applications, including cancer diagnosis. The diagnostic sensitivities of MMP-2, MMP-9, and fibronectin (FN1) were higher than those of other known biomarkers (CEA and CA19-9) currently used in clinical practice.

Besides, there is any impact on the immunity system?  About this you can read the articles listed below.  The text is quite long and, if possible, some other explanatory could be useful.

Response: We added and discussed the ECM immune functions.